# Adenine as a Halogen Bond Acceptor: A Combined Experimental and DFT Study

**Yannick Roselló [1], Mónica Benito [2,*], Elies Molins [2] , Miquel Barceló-Oliver [1] and Antonio Frontera [1,*]**

[1] Departament de Química, Universitat de les Illes Balears, Ctra. de Valldemossa km 7.5, 07122 Palma de Mallorca, Baleares, Spain; ynkrm11@gmail.com (Y.R.); miquel.barcelo@uib.es (M.B.-O.)

[2] Institut de Ciència de Materials de Barcelona (ICMAB-CSIC), c/ Til·lers, s/n, Campus UAB, 08193 Bellaterra, Spain; elies.molins@icmab.es

[*] Correspondence: mbenito@icmab.es (M.B.); toni.frontera@uib.es (A.F.)

**Abstract:** In this work, we report the cocrystallization of $N^9$-ethyladenine with 1,2,4,5-tetrafluoro-3,6-diiodobenzene (TFDIB), a classical XB donor. As far as our knowledge extends, this is the first cocrystal reported to date where an adenine derivative acts as a halogen bond acceptor. In the solid state, each adenine ring forms two centrosymmetric H-bonded dimers: one using N1···HA6–N6 and the other N7···HB6–N6. Therefore, only N3 is available as a halogen bond acceptor that, indeed, establishes an N···I halogen bonding interaction with TFDIB. The H-bonded dimers and halogen bonds have been investigated via DFT (Density Functional Theory) calculations and the Bader's Quantum Theory of Atoms In Molecules (QTAIM) method at the B3LYP/6-311+G* level of theory. The influence of H-bonding interactions on the lone pair donor ability of N3 has also been analyzed using the molecular electrostatic potential (MEP) surface calculations.

**Keywords:** $N^9$-ethyladenine; crystal engineering; noncovalent interactions; halogen bonding; hydrogen bonding; DFT; QTAIM; cocrystal; TFDIB

---

## 1. Introduction

Non-covalent interactions are crucial in regulating how molecules interact with one another [1]. Hydrogen bonding (HB) is often the predominant interaction and a common tool in supramolecular chemistry and crystal engineering [2]. In recent years, σ-hole interactions are new players in these fields [3,4]. A σ-hole is usually defined as an electropositive potential along the vector of a covalent bond. It is located at the σ* orbital of that bond [5,6]. Such σ-holes are commonly found on non-metals such as tetrel, pnicogen, and chalcogen and halogen atoms [5,7,8]. Interactions involving halogens are known as halogen bonding (XB) [9]. The first evidence of this occurrence was reported in 1954 in the crystal structure of 1,4-dioxane·$Br_2$ [10] and later highlighted by Hassel in his 1969 Nobel lecture. The halogen-bonding interaction has been used in supramolecular chemistry, catalysis, and crystal engineering [9]. Moreover, the biological significance of halogen bonding was demonstrated in 2004 [11], and a systematic analysis of the protein databank (PDB) [12] evidenced that X bonding is similar to H bonding from a structural perspective [13–15]. In addition, X bonding has been exploited in systematic drug discovery [16,17] and used to regulate the conformation of a four-stranded DNA junction [18] and to influence the conformation of peptides [19].

In this work, we selected $N^9$-ethyladenine since we were interested in investigating its halogen bonding capability and the competition with hydrogen bonding. In a biochemistry sense, adenine derivatives are valuable compounds. In addition, from a crystal engineering standpoint, they are interesting due to the presence of various potential donor and acceptor sites. In spite of this potential

interest, a search in the Cambridge Structural Database (CSD) [20] version 5.39 (last updated August 2018) revealed that there is not any X-ray structure comprising N[9]-substituted adenine as a halogen bond acceptor in the database. In this manuscript, we report the cocrystallization of N[9]-ethyladenine with 1,2,4,5-tetrafluoro-3,6-diiodobenzene (TFDIB). The solid-state structure of this cocrystal shows that adenine has a strong ability to form self-assembled and centrosymmetric H-bonded dimers where only N3 atoms (see Figure 1) remain available to establish X-bonding interactions. The ditopic TFDIB (Figure 1) is then able to connect the infinite 1D assemblies generated by the H-bonding network. We have used DFT calculations to analyze and compare the competitive H-bonding and X-bonding interactions, and also the influence of the H bonds on the X bond acceptor ability of N3.

**Figure 1.** Molecular structures of 1,2,4,5-tetrafluoro-3,6-diiodobenzene (TFDIB, left), N[9]-ethyladenine (middle), and an H-bonding network with indication of the two possible centrosymmetric H-bonds (right).

## 2. Materials and Methods

### 2.1. Materials

N[9]-ethyladenine was prepared by treating adenine with bromoethane using NaH as base, following the procedure described for the preparation of N[9],N[9']-trimethylenebisadenine [21]. Reagents used to prepare N[9]-ethyladenine were p.a. grade and used without further purification, as received from Sigma-Aldrich (Merck KGaA, Darmstadt, Germany) and Alfa Aesar (Massachusetts, USA). 1,2,4,5-tetrafluoro-3,6-diiodobenzene (TFDIB) was purchased from Cymit Química S. L. (Barcelona, Spain). Milli-Q water (Millipore, Merck KGaA, Darmstadt, Germany, conductivity of 18.2 MΩ cm at 25 °C and 1 ppb TOC) and analytical grade acetonitrile from Sigma-Aldrich (Merck KGaA, Darmstadt, Germany) were used for the crystallization experiments.

### 2.2. Synthesis of the Cocrystal 1

The cocrystal was prepared by mechanochemical synthesis as follows: a mixture of 2:1 stoichiometric molar ratio of N[9]-ethyladenine and 1,2,4,5-tetrafluoro-3,6-diiodobenzene (TFDIB) was ground for 30 min at 30 Hz in the presence of two drops of water using a Retsch mixer mill MM400 (Retsch GmbH, Haan, Germany) (in a 10 mL agate grinding jar with two 5 mm agate balls).

Slow evaporation of a solution of the previous material in acetonitrile afforded suitable single crystals in a couple of days.

### 2.3. Characterization

*Powder X-Ray Diffraction* (PXRD). X-ray powder patterns (Cu K$_\alpha$ radiation, step: 1°/s; step size: 0.02°; 45 kV; 35 mA) were collected in the 2θ range 2–50° using a Siemens D5000 diffractometer (Bruker Española, Madrid, Spain).

*Single Crystal X-Ray Diffraction* (SCXRD). A suitable crystal of the cocrystal **1** was selected for X-ray single crystal diffraction experiments and mounted at the tip of a glass fiber on an Enraf-Nonius

CAD4 diffractometer (Bruker Española, Madrid, Spain) producing graphite monochromated Mo $K_\alpha$ radiation ($\lambda$ = 0.71073 Å). After the random search of 25 reflections, the indexation procedure gave rise to the cell parameters. Intensity data were collected in the $\omega$-$2\theta$ scan mode and corrected for Lorenz and polarization effects at 294(2) K. The absorption correction was performed following the DIFABS [22] method. The structural resolution procedure was made using the WinGX package [23]. Solving for structure factor phases was performed by SIR2014 [24] and the full matrix refinement by SHELXL-2014/7 [25]. Non-H atoms were refined anisotropically and H atoms were introduced in calculated positions and refined riding on their parent atoms. The program Mercury was used to calculate the X-ray powder pattern from single crystal structure data [26].

Supplementary crystallographic data for this paper have been deposited at the Cambridge Crystallographic Data Centre (CCDC 1903339) and can be obtained free of charge via www.ccdc.cam.ac.uk/data_request/cif. Details about X-ray data quality and resolution can be found in Table 1.

**Table 1.** Crystallographic data and refinement for cocrystal **1**.

| Crystal | Cocrystal 1 |
|---|---|
| Empirical Formula | $C_{20}H_{18}F_4I_2N_{10}$ |
| $Mr$ | 728.24 |
| Crystal system | Triclinic |
| Space group | P $\bar{1}$ |
| a/Å | 5.490(8) |
| b/Å | 8.505(5) |
| c/Å | 14.446(8) |
| $\alpha$/° | 79.20(5) |
| $\beta$/° | 80.81(7) |
| $\gamma$/° | 73.29(7) |
| V/Å$^3$ | 630.6(11) |
| Z | 1 |
| Radiation type | Mo $K_\alpha$ |
| $\mu$/ mm$^{-1}$ | 2.554 |
| Temperature/K | 294(2) |
| Crystal size/ mm | 0.360 × 0.270 × 0.180 |
| $D_{calc}$/ g·cm$^{-3}$ | 1.918 |
| Reflections collected | 2320 |
| Independent Reflections | 2208 [R(int) = 0.0548] |
| Completeness to theta = 24.996° | 99.5 % |
| F(000) | 350 |
| Data/ restraints/ parameters | 2202/0/164 |
| Goodness-of-fit | 1.100 |
| Final R indices [I < 2d(I)] | R1 = 0.0507, wR2 = 0.1383 |
| R indices (all data) | R1 = 0.0738, wR2 = 0.1493 |
| Largest diff. peak and hole/ e·Å$^{-3}$ | 0.728 and −0.701 |
| CCDC number | 1903339 |

*Attenuated Total Reflection Fourier Transform Infrared (ATR-FT-IR) Spectroscopy.* Infrared spectra were recorded with a Jasco 4700LE spectrophotometer (Jasco Analitica Spain S.L., Madrid, Spain) with attenuated total reflectance accessory. The scanning range was 4000 to 400 cm$^{-1}$ and at a resolution of 4.0 cm$^{-1}$.

*Thermogravimetric Analysis–Differential Scanning Calorimetry (TGA-DSC).* Thermal analyses were carried out on a simultaneous thermogravimetric analysis (TGA)–differential scanning calorimetry/differential thermal analysis (heat flow DSC /DTA) system NETZSCH-STA 449 F1 Jupiter (Netzsch-Gerätebau GmbH Spain branch office, Barcelona, Spain). An accurately weighted sample was placed in an alumina pan and measured at a scan speed of 10 °C min$^{-1}$ from ambient temperature to 250 °C under $N_2$ atmosphere as protective and purge gas (their respective flow velocities were 20 and 40 mL/min).

### 2.4. Computational Details

The geometries of the complexes studied herein were initially retrieved from the experimental X-ray coordinates, which were fully optimized using DFT calculations. In particular, the level of theory selected for this work was the utilization of B3LYP functional [27] in combination with Grimme's dispersion correction [28] and the 6-311+G* basis set [29] by means of the Gaussian-09 [30] program package. For the iodine atom, we used the LANL2DZ basis set that uses effective core potentials [31]. The molecular electrostatic potential (MEP) surfaces were computed at the same level of theory and using the 0.001 a.u. isosurface as a good estimate of the van der Waals surface. The Gaussian-09 [30] program package was used to compute the MEP and Gaussview was used to generate the MEP surface plots. The Quantum Theory of Atoms in Molecules QTAIM [32] analysis (topological analysis of the electron density) was performed using the AIMall program [33] at the same level of theory. The Cartesian atomic coordinates are included in the supporting information file (Table S1).

## 3. Results and Discussion

### 3.1. Structural Analysis

Comparison of the experimental pattern for the solid obtained after grinding and the simulated pattern from the single crystal structure resulted in good agreement, confirming that cocrystal **1** could be obtained by both methods, grinding and evaporative crystallization (see Figure S1 in Supplementary Materials). It was further analyzed by ATR-FT-IR (Figure S2). It was expected that hydrogen and halogen bond formation would induce small changes in the associated bands of the participating molecules. For N$^9$-ethyladenine, an upshift in the corresponding bands for N-H and C=N from 1669, 1595, and 1571 cm$^{-1}$ (free) to 1673, 1604, and 1572 cm$^{-1}$ in cocrystal is observed. In the case of the TFDIB, the IR active $\nu_{C-C}$ and $\nu_{C-F}$ stretch modes are slightly upshifted from 1456 and 937 cm$^{-1}$ (free) to 1459 and 939 cm$^{-1}$ in the cocrystal. These small changes might be assigned to the intermolecular interactions between molecules.

The adduct N$^9$-ethyladenine–TFDIB cocrystallizes in the monoclinic P$\bar{1}$ space group containing one molecule of N$^9$-ethyladenine and half of a 1,2,4,5-tetrafluoro-3,6-diiodobenzene (TFDIB) in the asymmetric unit. No solvent molecules are observed, which confirms the results observed in the TGA-DSC traces (see Figure S3). No loss on drying appears in the TGA trace before melting and degradation. Moreover, in the DSC, only one endothermic peak is observed corresponding to the melting process with a T$_{peak}$ of 209.2 °C. This value is higher than the melting temperatures of the pure precursors, N$^9$-ethyladenine or TFDIB (190.8 °C and 107–109 °C, respectively). A representative ORTEP (Oak Ridge Thermal Ellipsoid Plot Program) diagram is shown in Figure 2.

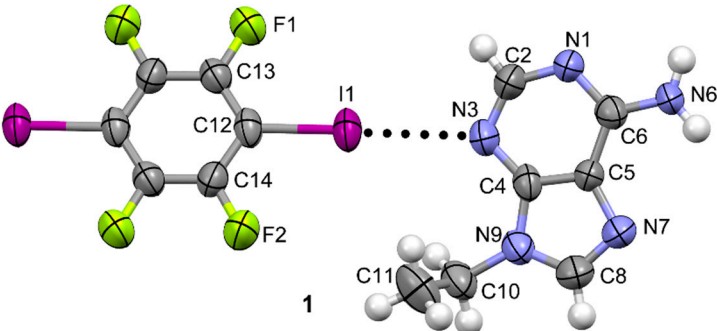

**Figure 2.** ORTEP plot of the cocrystal **1** with indication of the atom numbering scheme. Ellipsoids drawn at 50% probability level.

In this structure, N$^9$-ethyladenine molecules aggregate creating infinite 1D zig-zag tapes as a consequence of the formation of self-complementary H-bonding interactions (see Figure 3), both

through the Watson–Crick edge (N6···N(1) distance: 2.166 Å; angle 167.54°; red dotted lines) and the Hoogsteen edge (N6···N(7) distance: 2.196 Å; angle 159.68°; black dotted lines). Interestingly, TFDIB links $N^9$-ethyladenine zig-zag tapes by establishing two symmetrically equivalent halogen-bonding interactions (N(3)···I(1) distance: 2.939 Å; angle 175.28°; blue dotted lines).

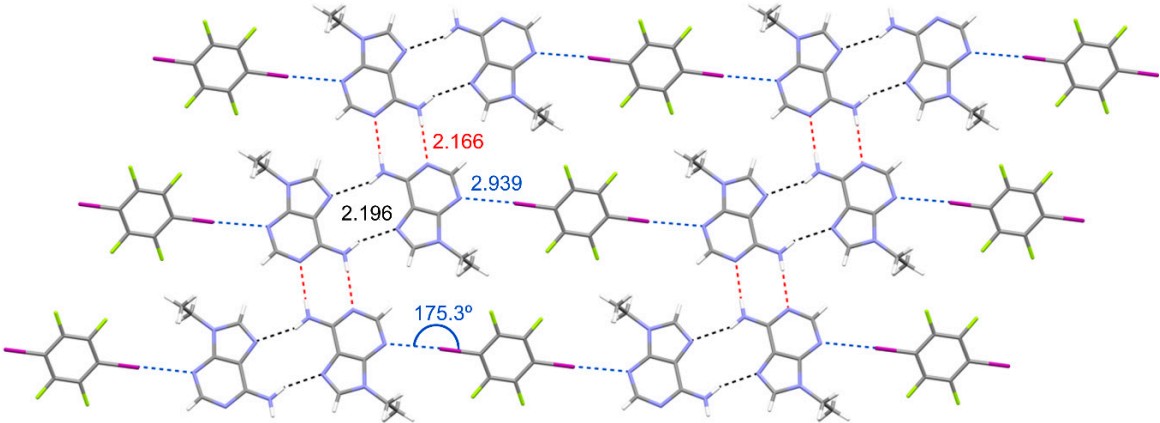

**Figure 3.** $N^9$-ethyladenine–TFDIB cocrystal planes formed by zig-zag $N^9$-ethyladenine tapes linked by TFDIB. Distances in Å.

As shown in Figure 4, the final 3D architecture of this compound is generated by the stacking of the 2D supramolecular planes described above (see Figure 3). It is worth mentioning that in addition to the π–π stacking interactions involving the adenine moieties (see Figure 4, left), lp–π interactions are also established between the negative belt of the iodine atom and the π-system of $N^9$-ethyladenine (see Figure 4, right). This type of X···π interaction where the negative belt of the halogen interacts with an acidic π-system has been previously described and discussed in the literature [34,35]. The I···π interaction is represented in Figure 4 as green dotted lines. The distance between the centroid (Cg) of the six-membered ring and iodine is 3.696 Å and the C–I···Cg angle is 100.38°.

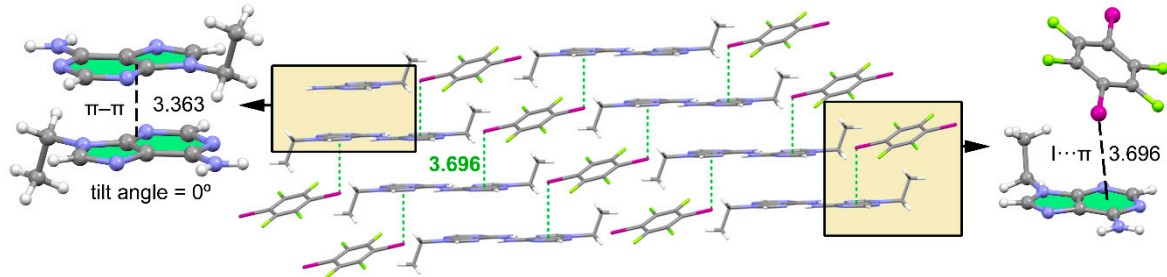

**Figure 4.** Partial view of the X-ray structure of **1** (middle) with indication of the π–π stacking (left) and lp–π interaction (right). Distances in Å. The π–π distance is measured between the mean planes of the adenine rings, which are parallel. The lp–π distance is measured from iodine to the ring centroid of the six-membered ring.

*3.2. Packing Efficiency*

To determine the efficiency of packing, we have determined the packing coefficient $C_K$ [36]. Each molecule in the crystal structure occupies a volume $V_{mol}$ that can be estimated using the van der Waals radii of its constituent atoms. Therefore, the packing coefficient for a crystal structure with a unit cell volume $V_{cell}$ is $C_K = (1/V_{cell}) \times \Sigma V_{mol}$; where $\Sigma V_{mol}$ is the sum of the volumes occupied by the molecules within the cell. We have calculated the packing coefficient of cocrystal **1**, $N^9$-ethyladenine (Refcode ZZZGGK01), and TFDIB (Refcodes: ZZZAVM01 and ZZZAVM02) using the PLATON package [37]. The change in packing coefficient upon forming the cocrystal is defined as the difference between the

packing coefficient ($C_K$) in the cocrystal and the stoichiometrically normalized sum of $C_K$ values of the single components: $\Delta C_K = C_K(X_n \cdot Y_m) - [C_K(X_n) + C_K(Y_m)]/(m + n)$.

Frequently, cocrystals are thermodynamically more stable than their single components. This can be related to an improvement in the packing efficiency upon formation of the cocrystal since it correlates with an improvement in the interactions in general. The packing efficiencies are 0.667 for **1**, 0.670 for $N^9$-ethyladenine [38], and 0.679 and 0.721 for polymorphs I [39] and II [40] of TFDIB, respectively. The resulting $\Delta C_K$ values are thus negative, viz –0.0075 and –0.033 using polymorphs I and II, respectively, of TFDIB. Therefore, these values reveal that the cocrystal packs less efficiently than their individual counterparts. This result strongly agrees with a recent report [41] that concluded that the energetic drivers of cocrystallization are more complex than simply improving the packing of the crystal. As a matter of fact, in the analysis of 350 halogen-bonded and hydrogen-bonded cocrystal structures, 75% of them presented negative values of $\Delta C_K$s [41]. As concluded by the original authors, this is due to the fact that the directionality of the strong, specific interactions such as hydrogen and halogen bonds used to build the cocrystals enforces their structures to sacrifice close packing to improve the directionality of these contacts.

### 3.3. Theoretical Study

The theoretical study was devoted to analyzing the H-bonding interactions observed in the solid state of **1** via the Watson–Crick and Hoogsteen edges and how they affect the ability of N3 to participate in halogen-bonding interactions.

We first computed the MEP surfaces of isolated $N^9$-ethyladenine, two H-bonded dimers (denoted as DIM1 and DIM2), and the trimer represented in Figure 1, with the purpose to compare the MEP values at the N1, N3, and N7 atoms. The MEP surfaces are shown in Figure 5, and some interesting issues can be extracted from the results. That is, the MEP values at the three N atoms of the adenine ring (see Figure 5a) are similar, being slightly more negative at N1 (see Figure 2 for the atomic numbering scheme), thus suggesting that N1 is the more basic (better lone pair donor). The MEP surface of DIM1 (Hoogsteen dimer, see Figure 5b) indicates that upon formation of this dimer, the MEP value at N1 becomes more negative (−33.7 kcal/mol), thus, increasing its basicity. In contrast, the MEP value at N3 is almost unaltered (0.3 kcal/mol more positive). This indicates that upon formation of the Hoogsteen dimer, the ability of N1 to participate in H-bonding interactions increases and consequently the formation of the Watson–Crick would be favored. Quite remarkably, the MEP surface of DIM2 (Watson–Crick dimer, see Figure 5c) reveals a similar behavior. Namely, upon dimer formation, the MEP value at N7 becomes more negative (−32.1 kcal/mol) compared to isolated $N^9$-ethyladenine and the MEP value at N3 is slightly more positive (0.5 kcal/mol). Therefore, the formation of the Watson–Crick dimer increases the ability of N7 to establish H-bonding interaction. Taken together, these results suggest that the formation of the supramolecular tape is cooperative. Finally, we have also computed the MEP surface of the trimer (Figure 5d) to investigate whether the basicity of N3 in the trimer increases, thus favoring the formation of the X-bonding interaction. The MEP value at N3 in the trimer (−30.6 kcal/mol) is similar to those in the dimers (−30.7 kcal/mol, −30.5 kcal/mol) but more positive than in the monomer (−31.0 kcal/mol). This likely indicates that there is no favorable cooperativity between the H bonds and the halogen bonds in the crystal structure.

We have optimized the dimers and trimer shown in Figure 6 and computed their interaction energies. Moreover, we have also obtained the Atoms in Molecules (AIM) distribution of bond critical points and bond paths to further analyze the cooperativity features commented on above. A bond path in combination with a bond critical point connecting two atoms is a clear indication of interaction [42]. It can be observed that the Watson–Crick dimer is 2.2 kcal/mol more stable than the Hoogsteen dimer. This agrees well with the values of ρ(r) and $\nabla^2$ρ(r) at the bond critical point (CP) that characterize the H-bonding interaction (small green spheres in Figure 6) and that are higher in the former (values in italics). It is well known that the values of ρ(r) and $\nabla^2$ρ(r) at the bond CP can be conveniently used as a measure of the strength of the interaction [43]. Quite remarkably, the interaction energy of the trimer

($\Delta E_3 = -27.9$ kcal/mol) is 0.5 kcal/mol more favorable than the sum of the two dimers ($\Delta E_1 + \Delta E_2$), thus suggesting a favorable cooperativity and agrees well with the formation of the infinite 1D tape in the solid state of the cocrystal. The distribution of bond CPs and bond paths in the trimer reveals an H···H interaction in the trimer that is established between C–H groups from the six- and five-membered rings of two adenine moieties. The interaction energy of the halogen bonded complex (Figure 6d) is moderately strong ($\Delta E_4 = -4.8$ kcal/mol), thus revealing that the H bonds are stronger than the X bonds in this particular system. The distribution of bond CPs and bond paths in this dimer suggests the existence of some ancillary C–H···I interactions involving the H atoms of the ethyl groups, which are expected to be very weak. Previous studies have reported the relevance of C–H···I interactions in the structure and properties of triiodobenzenes [44] and 1-methylpyridinium iodide derivatives [45]. The values of $\rho(r)$ and $\nabla^2\rho(r)$ at the bond CP that characterize the X-bonding interaction are smaller than those at the bond CPs of the H bonds in the dimers (Figure 6a,b), in line with the energetic results.

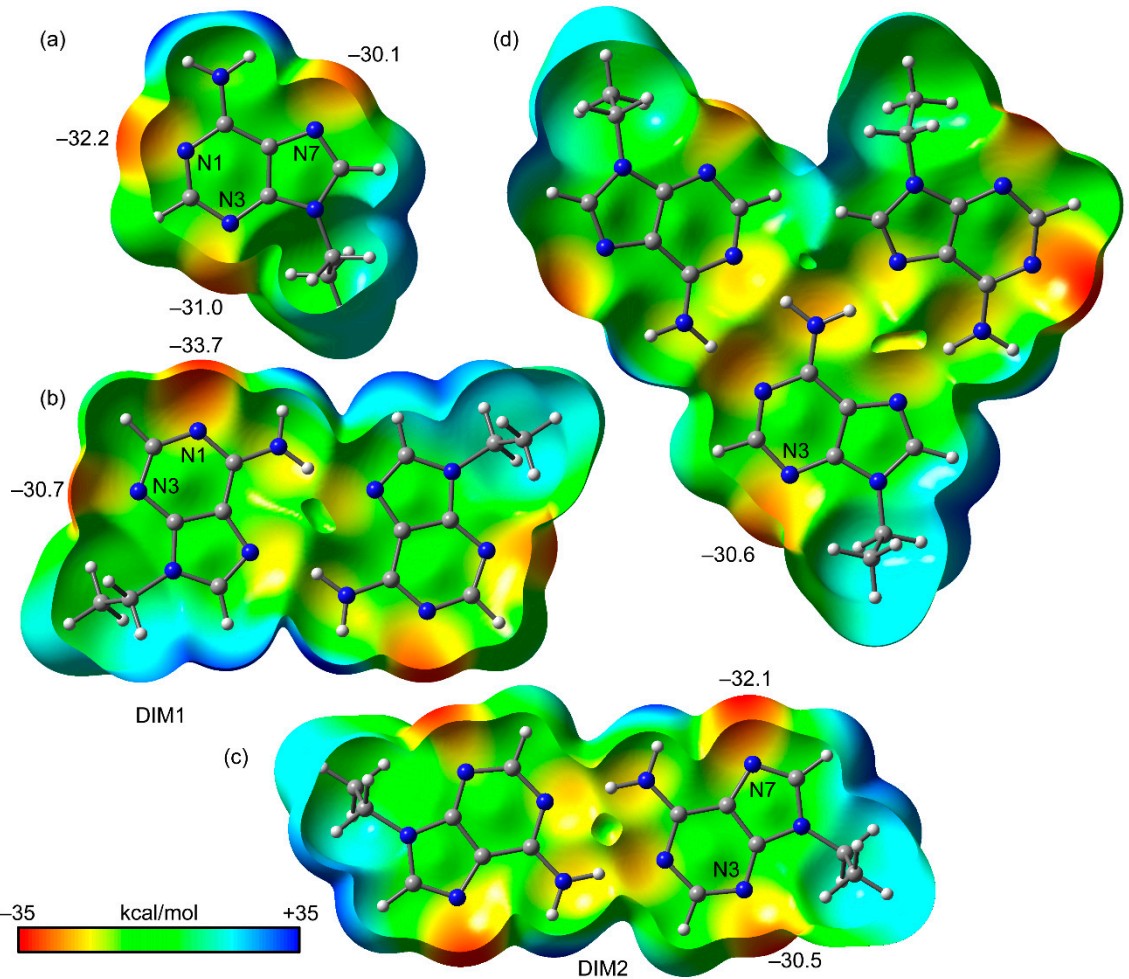

**Figure 5.** Open molecular electrostatic potential (MEP) surfaces (0.001 a.u. isosurface) of $N^9$-ethyladenine (**a**), DIM1 (**b**), DIM2 (**c**), and the trimer (**d**). The MEP values at selected points of the surface are indicated in kcal/mol.

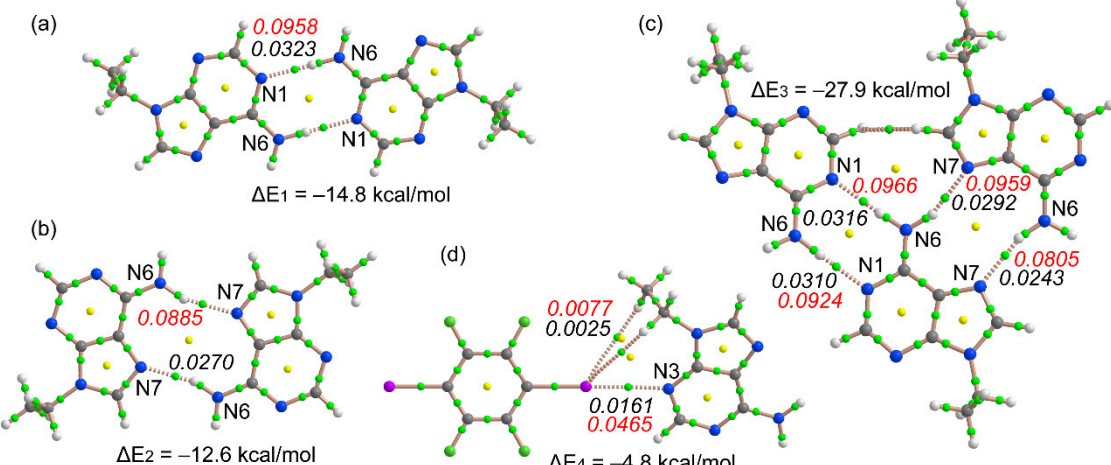

**Figure 6.** Distribution of bond and ring critical points (green and yellow spheres, respectively) and bond paths in the DFT-D optimized Watson–Crick dimer (**a**), the Hoogsteen dimer (**b**), the trimer (**c**), and the X-bonded dimer (**d**). The values of $\rho(r)$ and its Laplacian $[\nabla^2\rho(r)]$ at the bond critical points are indicated in italics (a.u.) using black and red font, respectively.

## 4. Conclusions

We have cocrystallized $N^9$-ethyladenine with a classical X-bond donor (1,2,4,5-tetrafluoro-3,6-diiodobenzene). The H bonds and X bonds coexist. In this sense, the adenine molecules form an H-bonding network in the solid state generating infinite 1D chains and these are interconnected via N3 by X-bonding interactions. This type of X bonding has not been observed for adenine before. By means of DFT energetic calculations, the QTAIM study, and the analysis of the MEP surfaces, we demonstrate cooperativity effects in the H-bonding interactions that justify the formation of the infinite 1D tape, leaving only N3 available for establishing X-bonding interactions that interconnect the tapes to form 2D planes.

**Supplementary Materials:** The following are available online at http://www.mdpi.com/2073-4352/9/4/224/s1, Figure S1: PXRD patterns of pure $N^9$-ethyladenine and 1,2,4,5-tetrafluoro-3,6-diiodobenzene (TFDIB) and experimental and simulated cocrystal **1**; Figure S2. Attenuated total reflection Fourier transform infrared (ATR-FT-IR) spectroscopy spectra of pure $N^9$-ethyladenine, 1,2,4,5-tetrafluoro-3,6-diiodobenzene (TFDIB), and cocrystal **1**; Figure S3. Thermogravimetric analysis–differential scanning calorimetry (TGA–DSC) trace of cocrystal **1**; Table S1. Cartesian coordinates of optimized complexes.

**Author Contributions:** Y.R. and M.B.-O. prepared the $N^9$-ethyladenine. Y.R. synthesized the cocrystal by grinding. M.B. designed the experiments and obtained the single crystal. E.M. performed the data collection and the resolution of the crystal structure. A.F. performed the D.F.T. study. M.B., M.B.-O., and A.F. designed the concept. All wrote and revised the manuscript.

**Funding:** This research was funded by MINECO/AEI from Spain, project numbers ENE2015-63969, CTQ2017-85821-R FEDER and SEV2015-0496.

**Acknowledgments:** We thank R. Frontera from the Centre de Tecnologies de la Informació (CTI) at the UIB for computational facilities.

**Conflicts of Interest:** The authors declare no conflict of interest.

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
