# Peer review of "Adenine as a Halogen Bond Acceptor: A Combined Experimental and DFT Study"

_crystals, doi:10.3390/cryst9040224_

Round 1

Reviewer 1 Report

The manuscript by Yannick Roselló, Mónica Benito, Elies Molins, Miquel Barceló-Oliver and Antonio Frontera,* entitled ''Adenine as a Halogen Bond Acceptor: A Combined Experimental and DFT Study'' reports a ‘’single’’ X-ray crystal structure of Adenine and tetrafluorodiiodobenzene with the focus on halogen-bonding features in the co-crystal. Halogen bond, an interaction that recently gained the attention it deserves, demonstrates its importance in various settings are research areas and proved to be a very useful tool in organic crystal engineering. The idea to explore a new set of halogen-bond acceptors in such systems studied here are very welcome and of substantially broad interest. Unfortunately, the presentation of the results, which certainly do have the potential to be published in the Crystals. I suggest a rejection of the manuscript, and its potential resubmission after a considerable revision is performed. 

Many terms used need rewording in sections, introduction, and results and discussions. Examples: ‘Line 28, 46,..’’ 

Some imprecise words and unclarified points:

Line 97: ‘’Accurately weighed sample....?’’ How much is accurate? Please provide amounts, mmoles, etc. 

Line 116-118: Comparison of the experimental pattern for the solid obtained after grinding and the simulated pattern from the single crystal structure resulted in good agreement, confirming that cocrystal 1 could be obtained by both methods (see Figure S1). 

Why did not authors simulate the solids obtained from mechanochemical synthesis with SCXRD? Why did they choose the grinding method to produce solids for PXRD? Do ‘’both methods’’ mean grinding and mechanochemical protocols? Where is the PXRD of solids (obtained by mechanochemical synthesis) in ESI?

The adenine-based compounds are extensively studied for decades. The obtained computational MEP values (not interesting), therefore, are anticipated. This is due to the aromatization character and dimerization ability of adenine compounds. Therefore, my question is, did authors tried to obtained co-crystals in different acceptor-to-donor ratios (by mechanochemical synthesis)? If attempts were made, and success was only in case of 2:1 ratio of acceptor:donor, then I recommend authors to include packing efficiency of dimer and trimer adenines (cif files from CSD and packing efficiency values by PLATON can be easily obtained), and compare it co-crystals, Adenine+tetrafluorodiiodobenzene. This would be of extremely interesting to see the molecular packing behavior upon the inclusion of more donor molecules crystal lattice. This is also due to the fact that the pi-pi stacking and halogen/lone-pair-pi interactions played a key role to obtain co-crystal, Adenine+tetrafluorodiiodobenzene (Authors also mentioned this point).

What is N...I-C angle? From Figure 4, it seems, the N...I-C angle is not perfectly linear?

Also, I recommend authors to present Figure 2 such that the co-crystal is displaying halogen bonding interaction.

Author Response

First of all we would like to thank this reviewer for his/her careful reading of the manuscript, corrections and suggestions. We have revised the manuscript taking into consideration his/her suggestions and corrections, as detailed below:

Many terms used need rewording in sections, introduction, and results and discussions. Examples: ‘Line 28, 46,..’’ 

 ANSWER: The document has been carefully revised.

 Some imprecise words and unclarified points:

 Line 97: ‘’Accurately weighed sample....?’’ How much is accurate? Please provide amounts, mmoles, etc. 

 ANSWER: the amount of sample used for this analysis was 11.21618(3) mg, taking the uncertainty from the guidelines given in the Netzsch website. On the other hand, precise mass is only needed when estimating enthalpies, which are not included in this contribution.

 Line 116-118: Comparison of the experimental pattern for the solid obtained after grinding and the simulated pattern from the single crystal structure resulted in good agreement, confirming that cocrystal 1 could be obtained by both methods (see Figure S1). 

Why did not authors simulate the solids obtained from mechanochemical synthesis with SCXRD? Why did they choose the grinding method to produce solids for PXRD? Do ‘’both methods’’ mean grinding and mechanochemical protocols? Where is the PXRD of solids (obtained by mechanochemical synthesis) in ESI?

ANSWER: Figure S1 from the Supplementary Information shows the comparison of the experimental powder x-ray diffraction pattern obtained by mechanochemical synthesis (also known as grinding) versus the simulated pattern obtained from the single crystal resolution. The grinding method is a quite common, fast, facile and green method not only for salt but also for cocrystal screenings [Cryst. Growth Des. 2009, 9, 1106-1123].

The adenine-based compounds are extensively studied for decades. The obtained computational MEP values (not interesting), therefore, are anticipated. This is due to the aromatization character and dimerization ability of adenine compounds. Therefore, my question is, did authors tried to obtained co-crystals in different acceptor-to-donor ratios (by mechanochemical synthesis)? If attempts were made, and success was only in case of 2:1 ratio of acceptor:donor, then I recommend authors to include packing efficiency of dimer and trimer adenines (cif files from CSD and packing efficiency values by PLATON can be easily obtained), and compare it co-crystals, Adenine+tetrafluorodiiodobenzene. This would be of extremely interesting to see the molecular packing behavior upon the inclusion of more donor molecules crystal lattice. This is also due to the fact that the pi-pi stacking and halogen/lone-pair-pi interactions played a key role to obtain co-crystal, Adenine+tetrafluorodiiodobenzene (Authors also mentioned this point).

ANSWER: 1:1 and 2:1 ratio for N9-ethyladenine-TFDIB mixtures were studied. However, only the 2:1 ratio gave a pure cocrystal. We have included and commented the packing efficiency of the cocrystal and its comparison with N9-ethyladenine (cif file ZZZGGK01) and TFDIB (cif files ZZAVM01 and ZZZAVM02). A new section has been added in the manuscript (3.2) with the results.

On the other hand, it is not clear for us how to calculate the packing efficiency of a “dimer” or a “trimer” of adenine, since these are not crystal structures (i.e., not available in the CSD). The packing efficiency (initially defined by Kitaigorodsky) refers to a cell volume, which is only defined in periodic structures and not for groups of molecules, as the dimer and trimer are. Perhaps we are misunderstanding the referee, so we kindly request him/her to clarify this point.

What is N...I-C angle? From Figure 4, it seems, the N...I-C angle is not perfectly linear?Also, I recommend authors to present Figure 2 such that the co-crystal is displaying halogen bonding interaction.

ANSWER: We have added the value of the N...I-C angle in Figure 3. We have also changed Figure 2 in order to display the halogen bonding interaction. In the corrected manuscript, the C-I···N angle is also included in line 136.

Reviewer 2 Report

The Article describes a nice example of halogen bonding (XB) in an organic binary co-crystal. Most interesting is the fact that N9-ethyl adenine is a derivative of an important nucleic base. The presence of multiple hydrogen bonding (HB) and XB acceptor sites as well as two HB donor sites with the possibility of self-complementary HB and the biological context make this compound an interesting object to study in the presence of a bidentate XB donor (diiodo-perfluorobenzene).

However there are important aspects to revise.

In contrast to the introduction of the manuscript (page 1, lines 29-31) and according to literature, hydrogen bonding (HB) is not defined as a specific case of halogen bonding (XB)! The corresponding paper cited by the authors [REF 4: Politzer, P.; Murray, J. S.; Clark, T. Halogen bonding and other σ-hole interactions: A perspective. Phys. 232 Chem. Chem. Phys. 2013, 15, 11178-11189.] clearly attributes sigma hole bonding to atoms of the groups IV-VII, thus not including protons: “ A σ-hole bond is a noncovalent interaction between a covalently-bonded atom of Groups IV–VII and a negative site, e.g. a lone pair of a Lewis base or an anion. It involves a region of positive electrostatic potential, labeled a σ-hole, on the extension of one of the covalent bonds to the atom. The σ-hole is due to the anisotropy of the atom's charge distribution. Halogen bonding is a subset of σ-hole interactions.”

To my mind the statement (page 1, lines 26-27) that “In general, supramolecular chemistry and crystal engineering rely (not relies) on HB interactions…” is not necessarily true. I would rather say that “often HB is the predominant interaction and a common tool …” since many structures do not involve HB at all.

Page 1, line 44: instead of “donor and acceptor species” better say “donor and acceptor sites”.

In the introduction it should be mentioned why the N9-ethyl derivative of adenine has been chosen. If other derivatives or the parent adenine itself have not afforded any co-crystals it could be a very useful comment for the reader.

In the Results and Discussion part the first paragraph has to be revised. Only one synthetic method is mentioned (grinding of the solid starting compounds) but two methods are insinuated. (lines 116-118).

Lines 128-129: the involved atoms concerning the given interatomic distances have to be indicated. In this case d(H6A…N1 and H6B…N7, respectively). Since the positions of the H are calculated it would be more accurate to add the d(N…N).

Line 131: The C-I…N angle is missing. Since one of the characteristics of sigma hole bonding is its strong directionality and for XB the bond angles close to 180° it is crucial to indicate both, interatomic distance and bond angle !

Line 132: Figure 3 has to be cited earlier in the text (at least in line 128) since the description of the “dotted lines” corresponds to the Figure 3.

Line 138. Pi-pi stacking is commonly accepted. In contrast 1p-pi interactions between the “negative belt of the iodine atom and the pi-system of…” are less known. An appropriate reference should be cited. Furthermore the distance and tilt angles of the pi systems involved in pi-pi stacking should be given and commented. As far as I remember there are few examples discussed at least back to 2005.

Lines 194-195: I am not certain that one can conclude on solid evidence for the “existence” of C-H…I interactions which should be extremely weak. There is still a controversy on interatomic distance of C-H…F hydrogen bonding in crystal structures (up to which distance one can suggest attractive interactions, especially if the H positions are calculated). When proposing such a hypothesis the authors have to be careful with the formulation and need to discuss the calculated geometry and also cite corresponding references on C-H…I interactions or mention the absence of appropriate articles.

Line 202: In the conclusion, the authors claim that HB prevails over XB. I do not agree at all, since both types of interactions are co-existing in the crystal structure and all existing HB donor sites as well as all XB donor sites are “coordinated/occupied”.

Lines 207-208: I also do not understand the last sentence of the conclusion. In which way to the results function as empirical principles of sigma hole interactions?

The FTIR spectra shown in the supplementary information part have to be discussed in the article.

I couldn’t find any data of melting (or decomposition) temperatures which are usually measured in such studies and compared with the characteristics of the pure starting compounds. This should be done for a revised version.

Author Response

First of all, we would like to thank this reviewer for his/her careful reading of the manuscript, corrections and suggestions. We have revised the manuscript taking into consideration his/her suggestions and corrections, as detailed below:

The Article describes a nice example of halogen bonding (XB) in an organic binary co-crystal. Most interesting is the fact that N9-ethyl adenine is a derivative of an important nucleic base. The presence of multiple hydrogen bonding (HB) and XB acceptor sites as well as two HB donor sites with the possibility of self-complementary HB and the biological context make this compound an interesting object to study in the presence of a bidentate XB donor (diiodo-perfluorobenzene).

However, there are important aspects to revise.

In contrast to the introduction of the manuscript (page 1, lines 29-31) and according to literature, hydrogen bonding (HB) is not defined as a specific case of halogen bonding (XB)! The corresponding paper cited by the authors [REF 4: Politzer, P.; Murray, J. S.; Clark, T. Halogen bonding and other σ-hole interactions: A perspective. Phys. 232 Chem. Chem. Phys. 2013, 15, 11178-11189.] clearly attributes sigma hole bonding to atoms of the groups IV-VII, thus not including protons: “A σ-hole bond is a noncovalent interaction between a covalently-bonded atom of Groups IV–VII and a negative site, e.g. a lone pair of a Lewis base or an anion. It involves a region of positive electrostatic potential, labeled a σ-hole, on the extension of one of the covalent bonds to the atom. The σ-hole is due to the anisotropy of the atom's charge distribution. Halogen bonding is a subset of σ-hole interactions.”

To my mind the statement (page 1, lines 26-27) that “In general, supramolecular chemistry and crystal engineering rely (not relies) on HB interactions…” is not necessarily true. I would rather say that “often HB is the predominant interaction and a common tool …” since many structures do not involve HB at all.

ANSWER: We have changed these sentences as required by the referee. However, we would like to emphasize that in the conclusions of ref 3 (Tim Clark’s review), it is stated “the σ-hole can exist for both halogens and, surprisingly, for hydrogen as a hydrogen-bond donor. This means that the directionality of both halogen and hydrogen bonds has at least a significant electrostatic component and that it is not necessary to invoke effects such as donor–acceptor interactions to explain this directionality.

Page 1, line 44: instead of “donor and acceptor species” better say “donor and acceptor sites”.

ANSWER: We have amended this and now it reads ‘sites’ instead of ‘species’ (line 44).

In the introduction it should be mentioned why the N9-ethyl derivative of adenine has been chosen. If other derivatives or the parent adenine itself have not afforded any co-crystals it could be a very useful comment for the reader.

ANSWER: In the present work, we decided to start the study with this derivative. But in a future work, we have planned to study other modified adenines.

In the Results and Discussion part the first paragraph has to be revised. Only one synthetic method is mentioned (grinding of the solid starting compounds) but two methods are insinuated. (lines 116-118).

ANSWER: To clarify, both methods are included ‘grinding and evaporative crystallization’ (line 118).

Lines 128-129: the involved atoms concerning the given interatomic distances have to be indicated. In this case d(H6A…N1 and H6B…N7, respectively). Since the positions of the H are calculated it would be more accurate to add the d(N…N).

ANSWER: Both interatomic distances are included as indicated: N6 ···N(1) and N6 ···N(7). So Hs have been removed.

Line 131: The C-I…N angle is missing. Since one of the characteristics of sigma hole bonding is its strong directionality and for XB the bond angles close to 180° it is crucial to indicate both, interatomic distance and bond angle !

ANSWER: The C-I···N (angle 175.28°) is now included in the revised version, line 141 and in Figure 3.

Line 132: Figure 3 has to be cited earlier in the text (at least in line 128) since the description of the “dotted lines” corresponds to the Figure 3.

ANSWER: Now in the corrected version, it is cited in line 137, the first time it is talked about interactions.

Line 138. Pi-pi stacking is commonly accepted. In contrast 1p-pi interactions between the “negative belt of the iodine atom and the pi-system of…” are less known. An appropriate reference should be cited. Furthermore, the distance and tilt angles of the pi systems involved in pi-pi stacking should be given and commented. As far as I remember there are few examples discussed at least back to 2005.

ANSWER: As suggested, we have added to references for the lp(belt)–π interactions and also included geometric details of the π–π interaction

Lines 194-195: I am not certain that one can conclude on solid evidence for the “existence” of C-H…I interactions which should be extremely weak. There is still a controversy on interatomic distance of C-H…F hydrogen bonding in crystal structures (up to which distance one can suggest attractive interactions, especially if the H positions are calculated). When proposing such a hypothesis the authors have to be careful with the formulation and need to discuss the calculated geometry and also cite corresponding references on C-H…I interactions or mention the absence of appropriate articles.

ANSWER: Two references have been added that have previously described and studied C–H···I interactions. We have also changed the sentence to be more careful describing these interactions.

Line 202: In the conclusion, the authors claim that HB prevails over XB. I do not agree at all, since both types of interactions are co-existing in the crystal structure and all existing HB donor sites as well as all XB donor sites are “coordinated/occupied”.

ANSWER: This sentence has been amended and now it reads as follows: ‘The H-bonds and X-bonds coexist. In this sense, adenine molecules form a H-bonding network in the solid state generating infinite 1D chains and these are interconnected via N3 by X-bonding interactions.’

Lines 207-208: I also do not understand the last sentence of the conclusion. In which way to the results function as empirical principles of sigma hole interactions?

ANSWER: This sentence has been eliminated. With this sentence we tried to state that this manuscript could be used to parametrize force fields from the energetic and geometric features reported therein.

The FTIR spectra shown in the supplementary information part have to be discussed in the article.

ANSWER: The main changes in the FTIR spectra of the cocrystal and the parent compounds have been discussed in the revised version (see lines 119-124).

I couldn’t find any data of melting (or decomposition) temperatures which are usually measured in such studies and compared with the characteristics of the pure starting compounds. This should be done for a revised version.

ANSWER: In the previous version of the manuscript, this information was indeed included in the Supplementary Information. Now, in the revised version, these comments appear in the manuscript, line 123.

Round 2

Reviewer 1 Report

Frontera et al. report ''Adenine as a Halogen Bond Acceptor: A Combined Experimental and DFT 
Study'' Authors have done a good job to improve the manuscript. Therefore, I recommend this article to be published in crystals journal.